# Disturbed Glucose Metabolism in Rat Neurons Exposed to Cerebrospinal Fluid Obtained from Multiple Sclerosis Subjects

**DOI:** 10.3390/brainsci8010001

**Published:** 2017-12-21

**Authors:** Deepali Mathur, Eva María-Lafuente, Juan R. Ureña-Peralta, Lucas Sorribes, Alberto Hernández, Bonaventura Casanova, Gerardo López-Rodas, Francisco Coret-Ferrer, Maria Burgal-Marti

**Affiliations:** 1Department of Functional Biology, University of Valencia, 46100 Valencia, Spain; matdeepali@gmail.com; 2Multiple Sclerosis Laboratory, Department of Biomedicine, Prince Felipe Research Center, 46012 Valencia, Spain; zacapardo@gmail.com (E.M.-L.); jurena@cipf.es (J.R.U.-P.); lucas.sorribes@gmail.com (L.S.); alheca73@hotmail.com (A.H.); 3CSUR-Esclerosi Múltiple, Hospital Universitari i Politècnic La Fe, Unitat Mixta d’Esclerosi Múltiple i Neurorregeneració de l’IIS-La Fe, 46026 València, Spain; casanova.bonaventura@gmail.com; 4Department of Biochemistry and Molecular Biology, University of Valencia and INCLIVA Biomedical Research Institute, 46100 Valencia, Spain; gerardo.lopez@uv.es; 5Hospital Clínico, Universitario de Valencia, 46010 Valencia, Spain; coretf@gmail.com

**Keywords:** multiple sclerosis, glucose metabolism, neuromyelitis optica, cerebrospinal fluid, gene expression

## Abstract

Axonal damage is widely accepted as a major cause of permanent functional disability in Multiple Sclerosis (MS). In relapsing-remitting MS, there is a possibility of remyelination by myelin producing cells and restoration of neurological function. The purpose of this study was to delineate the pathophysiological mechanisms underpinning axonal injury through hitherto unknown factors present in cerebrospinal fluid (CSF) that may regulate axonal damage, remyelinate the axon and make functional recovery possible. We employed primary cultures of rat unmyelinated cerebellar granule neurons and treated them with CSF obtained from MS and Neuromyelitis optica (NMO) patients. We performed microarray gene expression profiling to study changes in gene expression in treated neurons as compared to controls. Additionally, we determined the influence of gene-gene interaction upon the whole metabolic network in our experimental conditions using the Search Tool for the Retrieval of Interacting Genes/Proteins (STRING) program. Our findings revealed the downregulated expression of genes involved in glucose metabolism in MS-derived CSF-treated neurons and upregulated expression of genes in NMO-derived CSF-treated neurons. We conclude that factors in the CSF of these patients caused a perturbation in metabolic gene(s) expression and suggest that MS appears to be linked with metabolic deformity.

## 1. Introduction

Multiple Sclerosis (MS) is an inflammatory demyelinating neurological disease that affects more than two million people globally. Both neurons and glia are regarded as the cellular “victims” in MS. Axonal damage has now been widely accepted as the major cause of permanent functional disability in patients with MS, with unknown origin. In relapsing-remitting MS subtype (RRMS), oligodendrocytes have the potential to remyelinate and repair the damaged axon and restore neurological function. Furthermore, cerebrospinal fluid (CSF) is closely juxtaposed with brain parenchyma and contains many proteins and molecules, which may impact the cellular physiology of brain cells. It is a promising biological fluid for discovering putative biomarkers and disease-associated proteins in MS, both with respect to inflammatory and neurodegenerative processes. Investigators have reported that neurotoxicity occurs in culture when xenogeneic models are exposed to human CSF, albeit the molecular pathological mechanisms remain elusive [1,2]. Our group reported that bioenergetics is impaired in rat neuronal cultures by ceramide molecules present in the CSF of MS patients [3]. It is well known that cells require copious amounts of energy when Na^+^/K^+^ ATPases are redistributed in demyelinated axons [4] and there is a disturbance in metabolic or trophic support due to damaged oligodendrocytes [5]. This is attributable to axonal injury, which causes irreversible permanent neurological deficits in MS patients. However, there are diffusible factors, hitherto unknown, present in the CSF of MS patients that may regulate axonal damage and repair during RRMS. 

Previous literature reveals that there is a metabolic dysfunction observed in patients suffering with neurodegenerative diseases [6,7,8,9,10,11,12,13,14,15,16,17,18,19,20,21,22,23,24,25]. Studies have shown that mitochondrial structure is altered in the motor neurons of mutant SOD1 mice, a model of Amyotrophic Lateral Sclerosis (ALS) [24,26,27,28]. Further, the activity of complex I of the electron transport chain in mitochondria is significantly reduced in SOD1G93A mice [29,30] and glucose metabolism, as well as ATP formation, is impaired in the spinal cords of mutant SOD1 mice, which indicates mitochondrial abnormalities [30,31]. The levels of cytochrome oxidase-1 and mitochondrial DNA significantly increased in hippocampal neurons, although the number of mitochondria per neuron declined in Alzheimer’s disease [32]. Overall, these reports suggest an alteration in metabolic pathways, oxidative imbalance and bioenergetics failure in neurodegenerative diseases. Several other studies determined gene expression in peripheral mononuclear white blood cells [33,34,35,36,37], in brain tissues [38,39,40] and in the CSF of MS patients [41]. Findings from our previous work demonstrated that glyceraldehyde-3-phosphate dehydrogenase (*Gapdh*), a commonly used housekeeping gene, revealed reduced expression when cerebellar granule neurons (CGNs) were treated with CSF obtained from distinct clinical types of MS and Neuromyelitis optica (NMO) patients [42]. Our group recently reported bioenergetics failure in rat oligodendrocyte progenitor cells (OPCs) when they were treated with CSF obtained from different clinical forms of MS and NMO patients [43]. We observed the downregulation of genes involved in carbohydrate metabolism and reduced ATP synthesis for repairing axonal damage by OPCs in MS- and NMO-derived CSF-treated OPCs. We hypothesized that there are factors in the CSF of MS patients that may alter the expression of metabolic genes and ATP production and that these transcriptional changes may be related with the differential restoration capacity of neuronal damage between MS subtypes. Identification of pathophysiological mechanisms involved in axonal degeneration-reconstruction may shed light in the understanding MS progression and/or prognosis.

Finally, we sought to identify potential biomarkers that could differentiate MS from the similar but different neurological disease NMO. NMO, also known as Devic’s disease, is defined as a severe monophasic syndrome characterized by bilateral optic neuritis, acute transverse myelitis and inflammatory demyelination of the central nervous system (CNS) that selectively affects the spinal cord and optic nerves, resulting in blindness and paralysis but spares the brain [44]. NMO was previously considered a variant of MS; however, investigations have demonstrated that NMO patient sera contains NMO-specific Immunoglobulin G (IgG) antibodies (IgG-antibodies) and that these patients require a different treatment approach [45]. These antibodies target aquaporin-4, which is a channel that regulates water entry and exit of the nerve cells in the CNS [46,47].

The goal of the present study was to determine the effect of CSF on CGNs, which could contain factors that regulate axonal damage. Secondly, we wanted to elucidate candidate biomarkers in various clinical forms of MS and NMO to target and monitor the effects of future therapies aimed at preventing MS disease progression.

## 2. Materials and Methods

### 2.1. Study Approval

All procedures were approved by the Committee of Animal Care of Prince Felipe Research Center (CIPF), Valencia, in accordance with the regulations of the European Union and Spanish legislation. Written informed consent was obtained from all the patients and controls for this study and authorized by the Ethical Committee of the Hospital Universitario y Politécnico La Fe and Hospital Clínico Universitario de Valencia for this research. The study was approved by the Health Research Fund of the Institute of Health Carlos III (R + D + I 2009–2012; PS09/00976).

### 2.2. Patient Cohort

#### 2.2.1. Patient Population

A total of 59 patients were recruited and CSF samples were obtained from the Department of Neurology, Hospital Universitario y Politécnico La Fe and Hospital Clínico Universitario de Valencia. Out of 59 patients, 21 had Relapsing MS (11 G+/M+ and 10 G+/M−), 8 had MS with a predominant spinal cord affectation (Spinal MS), 11 had primary progressive MS (PPMS), 9 had NMO and 10 were non-inflammatory neurological controls (NIND patients) (Table 1). In CSF, apart from factors related to MS or NMO, there are factors present due to other diseases that produce their action. This is considered the “background noise” of the average population. The mixing of total CSF samples in all clinical forms may potentiate the factors related to MS. Therefore, samples from patients suffering from the same form of MS (e.g., G+/M−, G+/M+, Spinal MS, PPMS, or controls) were pooled together, several CGN cultures were treated with the CSF mixtures and RNA was extracted from the cultures (see below). 

MS patients were defined and grouped in different clinical courses, according to the current criteria [48] and diagnosed according to the McDonald criteria. They all met the following characteristics: oligoclonal IgG bands (OCGB) were present, the patient was not in a phase of relapse and the patient had received the last dose of steroids at least one month prior. Wingerchuk criteria were used to diagnose patients with NMO disease [49]. Patients who suffered relapses of optic neuritis and myelitis, who exhibited only two of the three aforementioned criteria, a normal MRI, or that did not satisfy the Patty criteria for MRI diagnosis of MS were included in the study. Table 1 describes the clinical characteristics of the patients.

#### 2.2.2. Patient Characteristics

Relapsing MS (RRMS and SPMS forms): MS with presence of relapses with or without progression is categorized into: (1) Relapsing MS (RMS) when progression of disability is not clinically evident and secondary progressive stage MS (SPMS) when a sustained increase in disability was demonstrated and relapses were also present yet; and (2) primary progressive MS (PPMS). Over 95% of patients with MS exhibit oligoclonal bands (OCBs) of IgG in CSF (G+) [50] and 40% exhibit Inmunoglobulin M (IgM) OCBs in CSF (M+), which is related to a more aggressive course of disease [51]. In this study, we further classified Relapsing MS into “G+/M−” and “G+/M+” subtypes (see below) on the basis of aggressivity and prognosis, which provides more complete information than just RMS or SPMS. In addition, we have studied a set of MS patients with a predominant affectation of the spinal cord separately because these patients have some peculiarities and we wanted to explore whether they exhibited differences in light of our experiments. These patients exhibited a diffused spinal cord affectation on the Magnetic Resonance Imaging (MRI) studies, with an initial relapsing course and a rapid evolution to the progressive phase [52]. 

The clinical characteristics of patients were studied by the expanded disability status scale (EDSS) and the multiple sclerosis severity score (MSSS). This is a scale normalized to a European population that offers the evolution of disability related to the years of duration of the diseases since diagnosis. The MSSS was retrospectively calculated for the last visit; individual values were obtained from the intersection of the column corresponding to EDSS and the row corresponding to the number of years from the first MS event.

RMS G+/M− clinical form of MS: Patients named as “G+/M− subtype” had IgG antibodies (+) but no IgM (−) oligoclonal antibodies detected in the CSF of brain. RMS G+/M+ clinical form of MS: Patients named as “G+/M+ subtype” had both IgG antibodies (+) and IgM (+) oligoclonal antibodies detected in the CSF of brain. Spinal MS: These patients were positive for oligoclonal IgG bands (OCGBs) and negative for oligoclonal IgM bands (OCMBs) in CSF. These patients also satisfied Swanton´s criteria for dissemination in time. PPMS: These patients are characterized by progressive decline in neurological disability. Neuromyelitis optica (NMO) patients: Individuals that met at least two of the following three features: (1) Long extensive transverse myelitis (>3 vestibule bodies); (2) Antibodies against aquaporin-4; (3) Normal brain at the first event. Controls (Non-Inflammatory Neurological Diseases (NIND)): Individuals who were suspected to have MS but after protocolized analysis were not diagnosed with MS.

### 2.3. Cerebrospinal Fluid Samples of Patients

Cerebrospinal fluid (CSF) samples were obtained by lumbar puncture at the time of diagnosis. No patient had received treatment with immunosuppressive drugs, immunomodulators, or corticosteroids for at least one month prior to the extraction of CSF. The routine clinical extraction of CSF from patients was 10 mL obtained by lumbar puncture in subarachnoid space under sterile conditions every three to six months. The samples were centrifuged for 10 min at 700× *g* and aliquots were frozen and stored at −80 °C in 1 mL aliquots until use. To preserve the integrity of the samples, the aliquots were used just once in an experiment without re-freezing.

### 2.4. Animals

Eight-day-old Wistar rats (Harlan Iberica, Sant Feliu de Codines, Barcelona, Spain) weighing between 200–250 g were used. All animals were raised under controlled conditions with cycles of light/dark (12/12 h), temperature of 23 °C and humidity of 60%. Access to water and food (standard rodent feed supplied by Harlan (Harlan Ibérica, Sant Feliu de Codines, Barcelona, Spain), Teklad 2014 Global 14% Protein Rodent Maintenance Diet) was provided. The maintenance of the animals was performed in the animal facilities unit of Prince Felipe Research Center (CIPF), Valencia, Spain.

### 2.5. Primary Culture of CGNs

Primary cultures of CGNs were obtained following a modified protocol [53]. Forebrains were collected from eight days old Wistar rats and mechanically dissociated and the cerebellum was dissected. Isolated cerebella were stripped of meninges, minced by mild trituration with a Pasteur pipette and treated with 3 mg/mL dispase (grade II) for 30 min at 37 °C in a 5% CO_2_ humidified atmosphere. After half an hour, dispase was inactivated with 1 mM EDTA. Granule cells were then resuspended in Basal Medium Eagle (BME, Gibco, ref. 41010) with 40 µg/mL of DNaseI. The cell suspension was filtered through a mesh with a pore size of 90 µm and centrifuged at 800× *g* for 5 min and thereafter, the cell suspension was washed three times with BME. Finally, the cells were resuspended in complete BME medium with Earle’s salts containing 10% heat inactivated fetal bovine serum (FBS, Gibco), 2 mM glutamine, 0.1 mg/mL gentamycin and 25 mM KCl. The neuronal cells were counted and plated onto poly-L-lysine coated 6-well (35-mm) culture dishes (Fisher) at a density of 3 × 10^5^ cells/well. Cells were maintained in an incubator at 37 °C, in a 5% CO_2_ with 95% humidity atmosphere. After 20 min at 37 °C, the medium was removed and fresh complete medium was added. Then, 20 µL of cytosine arabinoside (1 mM) was added to each culture plate after 18 to 24 h to inhibit replication of non-neuronal cells. To check cell viability, the dead cells were marked with propidium iodode (PI) (Sigma-Aldrich-Merck KGaA, Darmstadt, Germany) and living cells with Rhodine-123 (Sigma-Aldrich-Merck KGaA, Darmstadt, Germany).

### 2.6. Confocal Microscopy

Living cells were maintained at 37 °C and 5% CO_2_. Cells were analyzed on a Leica TCS SP2 confocal microscope acousto- optical beam splitter (AOBS) (Leica Microsystems, Leica Microsystem, Barcelona, Spain) inverted laser scanning confocal microscope using a 63Å~ Plan-Apochromat-Lambda Blue 1.4 N.A. oil objective lens. All confocal images were obtained under identical scan settings. Images of 1024 Å, 1024 pixels, 8-bits were collected for each preparation. Best focus was based on highest pixel intensity. Imaging conditions were identical for all the images and no images were saturated. Metamorph 7.0 (Molecular Devices, Downingtown, PA, USA) was used for image analysis on the images collected.

### 2.7. CSF Treatment and RNA Extraction

The neuronal cell cultures were treated for 24 h with CSF 1:10 in BME, 10% FBS 2 mM glutamine, 0.1 mg/mL gentamycin and 25 mM KCl. RNA was extracted using Quick RNA MicroPrep Kit (Zymo Research Corp, Irvine, KY, USA), according to the manufacturer’s instructions. The dilution 1:10 of CSF in BME does not change cell viability as determined by routine microscopy after staining dead cells with propidium iodode (PI) and living cells with Rhodine-123. 

The RNA concentration from neurons was determined spectrophotometrically at 260 nm using the Nanodrop 1000 spectrophotometer (V3.7 software, Agilent Technologies, Santa Clara, CA, USA). The quality of every RNA sample was measured by absorbance ratio at 260/280 nm and by capillary electrophoresis using 2100 Bioanalyzer instrument (Agilent Technologies, Santa Clara, CA, USA). For microarray assays, we used samples with a minimum RNA Integrity Number (RIN) score of 9.8 (mainly 10) in the Total RNA Nano Series. Isolated RNA was stored at −80 °C and later subjected to one color microarray-based gene expression analysis (Agilent Technologies, Santa Clara, CA, USA).

### 2.8. Gene Microarray and Data Normalization 

The labeled cRNA was hybridized to the Agilent SurePrint G3R at GE 8 × 60 K Microarray (GEO-GPL13521, in situ oligonucleotide), according to the manufacturer’s protocol. Briefly, the mRNA was reverse transcribed in the presence of T7-oligo-dT primer to produce cDNA. cDNA was then in vitro transcribed with T7 RNA polymerase in the presence of Cy3-CTP to produce labeled cRNA. The labeled cRNA was hybridized to the Agilent Sure Print G3 Rat GE 8 × 60 K Microarray according to the manufacturer’s protocol. The arrays were washed and scanned on an Agilent G2565CA microarray scanner at 100% PMT and 3 μm resolution. The intensity data was extracted using the *Feature Extraction Software* (Agilent Technologies, Santa Clara, CA, USA). The 75th percentile signal value was used to normalize Agilent one-color microarray signals for inter-array comparisons. After normalization, the data were filtered in order to exclude probesets with low expression and/or affected by differences between the laboratories. 

Differentially expressed genes were identified by comparing average expression levels in cases and controls. To normalize the data of expression of genes related with the carbohydrate metabolism, we selected a panel of reference genes from microarray data and analyzed by *geNorm* and *NormFinder* algorithms. After the analysis, we find that *Tfrc* and *B2m* the most stably expressed genes in the treatment of CGNs with CSF of the different MS and NMO patients (see [42]). Those genes were used as endogenous controls to normalize the microarray data obtained.

After this normalization, the mRNA expression (in terms of absolute fold change) in neurons treated with CSF of MS and NMO individual patients was compared with gene expression in neurons exposed to CSF from neurological controls. Fold change cutoff was considered as 2. The microarray data correspond to at least 3–4 independent assay per group of individual MS patients and controls.

### 2.9. Statistical Analysis

Statistical analysis was conducted after background noise correction using the *NormExp* method. Differential expression analysis was carried out on non-control probes with an empirical Bayes approach on linear models. Results were corrected for multiple testing hypothesis using false discovery rate (FDR) and all statistical analyses were performed with the *Bioconductor 3.5 project* (http://www.bioconductor.org/) in the *R statistical 3.4 environment* (http://cran.r-project.org/). To filter out low expressed features, the 30% quantile of the whole array was calculated and probe sets falling below threshold were filtered out. After merging the probes corresponding to the same gene on the microarray, the statistical significance of difference in gene expression were assessed using a standard one-way Analysis of Variance (ANOVA) followed by Tukey’s HSD (Honestly-significant-difference) post hoc analysis (cutoff *p* < 0.01 and FDR < 0.1). All data processing and analysis including PCA (Principal Component Analysis) plot plot was carried out using R functions.

### 2.10. Analysis of Gene-Gene Interaction Networks Using String v10 Software

We used STRING v10 software (https://string-db.org/) and correlated the protein-protein interactions in different disease subtypes in CGNs [54]. We defined a parameter to “*integrate*” our data as “*Cumulative Flux Index* or *CFI*” within the network to compare our experimental MS conditions. 

The values mean that the reduction of local flux due to the inhibition of an enzymatic activity in a specific gene affected synergically to the whole metabolic flux network. It means that as more genes were down regulated and with more intensity, the total flux was reduced as a cumulative factor that we integrated as the total CFI of the network. 

## 3. Results

The baseline characteristics of the study population are described in Table 2. The Prevalence of MS was found more in women (75%) than in men. The mean age of MS patients studied was 30.7 ± 9.7 years, whereas the mean age for NMO patients was 25.6 ± 15 years. According to classical clinical classifications of RMS, SPMS and PPMS, which only identifies the general characteristics of MS patients, there were significant differences observed in age at the beginning of PPMS and the other two MS forms (*p* < 0.003), in the Expanded Disability Status Scale (EDSS) of RRMS and the two other MS forms (*p* < 0.001) and in the time from the first to the second episode between PPMS and RRMS (*p* = 0.043) (Table 3). In the patients experiencing a progressive course, the duration of the disease was similar in secondary progressive cases and in cases that were progressive from onset (13.5 vs. 13.8) (Table 3). 

In Table 4, we categorize MS patients by the presence of CSF-restricted IgM OCB associated with an active inflammatory disease phenotype in RMS patients with more active inflammatory disease. With this working classification, we found significant differences in the age at disease beginning between PPMS and the other two MS forms (RRMS and SPMS) (*p* < 0.003). The people with PPMS are usually older at the time of diagnosis, with an average age of 40, in comparison with other MS patients. 

The different subtypes of MS help to predict disease severity and response to treatment, hence their categorization may be important for the clinical management of patients with MS. Moreover, we found significant differences in EDSS between RRMS and the two other MS forms (SPMS and PPMS) (*p* < 0.001). In this sense and when analyzed the MSSS, the highest values were for the spinal MS, followed by the PPMS and the RMS, being more aggressive patients with the G+/M+ phenotype (*p* < 0.001). 

Although nerve injury always occurs, the pattern is specific for each individual with MS. It is also noteworthy that disease severity and disability increases from RMS to SPMS course, in PPMS subtype and mainly when spinal cord affectation is present. The symptoms continually worsen from the time of diagnosis rather than having well-defined attacks and recovery. 

### 3.1. CSF Exposure Caused Neuronal Cell Death

We found that neuronal cells underwent cell death as they incorporated the dye propidium iodide (red fluorescence). However, astrocytes remained viable and retained their mitochondrial membrane potential by incorporating rhodamine-123 dye (green fluorescence). Figure 1 shows healthy neurons and astrocytes in control sample. Dead neuronal cells and viable astrocytes were observed in MS-derived CSF-treated CGNs cultures. The results suggest that factors present in the CSF of MS patients may have caused cell death in cerebellar granule neurons but not in astrocytes. 

### 3.2. Differential Expression of Genes Involved in Glucose Metabolism in CGNs by Microarray Gene Expression Profiling

The genes related with carbohydrate metabolism were normalized using endogenous control expression signal of *Tfrc* and *B2m* genes, two stably expressed genes previously found in the treatment of CGNs with the CSF of the different MS and NMO patients [42]. 

Our findings revealed that genes involved in carbohydrate metabolism were differentially expressed in our experimental conditions. Figure 2, Figure 3 and Figure 4 shows gene expression involved in the glycolytic pathway, the tricarboxylic acid (TCA) cycle and oxidative phosphorylation in CGNs treated with CSF derived from MS and NMO patients. The absolute fold change values from microarray data obtained for the different pathological patients were normalized to gene expression in CGNs treated with CSF derived from non-inflammatory neurological controls. 

When CGNs were exposed to the CSF of G+/M− RMS patients, glycolytic genes such as *Gapdh, Pgam* and *Eno1* showed reduced expression (Figure 2B,D,E). Furthermore, genes implicated in the TCA cycle, including *Pdha1* and *Mdh2,* were downregulated (Figure 3A,B). Similarly, *ATP5b*, a gene involved in oxidative phosphorylation, showed decreased expression as compared to neurological controls (Figure 4B).

When CGNs were exposed to the CSF of G+/M+ RMS patients, we observed strongly reduced gene expression of most of the enzymes involved in glycolysis, including *Hk1, Gapdh, Pgk1, Pgam and Eno1* (Figure 2A–E); the related TCA cycle enzymes, including *Pdha1* and *Mdh2* (Figure 3A,B); and the mitochondrial electron transport chain enzymes, such as *ATP5b*, when compared to neurological controls (Figure 4B).

Similarly, our findings revealed that expression of most of the enzymes involved in glycolysis, including *Hk1, Gapdh, Eno1* and *Pkm* (Figure 2A,B,E,F), were downregulated when CGNs were exposed to the CSF of spinal MS patients. Likewise, the expression of related TCA cycle enzymes, including *Pdha1 and Mdh2* (Figure 3A,B) and the mitochondrial electron chain enzymes (*ATP5a1* and *ATP5b*) were strongly reduced (Figure 4A,B).

The expression of the glycolytic gene *Pgam1* declined when neurons were treated with CSF derived from PPMS patients (Figure 2D). Our findings also revealed the downregulation of expression of genes implicated in the TCA cycle, including *Pdha1* and *Mdh2* (Figure 3A,B) and genes of the electron transport chain, such as *ATP synthase* (both alpha and beta subunits) (Figure 4A,B).

When CGNs were exposed to the CSF of NMO patients, most of the enzymes involved in glycolysis, including *Hk1, Pgam1 and Eno1* (Figure 2A,D,E), exhibited upregulated expression. The expression of the related TCA cycle enzymes, including *Mdh2* and *Aco2*, were downregulated, whereas expression of *Pdha1* was upregulated (Figure 3A,B). The mitochondrial electron chain enzymes (*ATP5a1* and *ATP5b*) were upregulated in gene expression (Figure 4A,B).

Overall, the microarray data demonstrate that the genes involved in carbohydrate metabolism were differentially expressed in CGNs treated with the CSF from MS and NMO patients compared to CGNs exposed to the CSF of neurological controls. We conclude that CSF exposure to CGNs altered the carbohydrate metabolism and may have altered the capacity of these cells to repair axonal damage in different clinical forms of MS. 

### 3.3. Analysis of Gene-Gene Interaction Networks Using String v10 Software

Figure 5 illustrates a schematic metabolic network including glycolysis, TCA cycle and electron transport chain with cumulative flux indexes. The gene-gene interaction network was visualized in CGNs exposed to the CSF from G+/M− MS patients (Figure 6A), G+/M+ MS patients (Figure 6B), spinal MS patients (Med) (Figure 6C), PPMS patients (Figure 6D) and NMO patients (Figure 6E) generated by STRING v10. The significantly downregulated genes were indicated by blue color and significantly upregulated genes were indicated by red color in the STRING figure. 

The variation of metabolic flux, estimated with their values of CFI, in the different treatments with CSF of MS and NMO patients were integrated. The CFI values were calculated as a parameter to integrate the reduced activity of the different enzymes in a specific network (glycolysis, TCA cycle and ATP generation, or together) as the expected total flux. This parameter is a simplified linear cumulative form of the flux control coefficients in the metabolic control analysis. This value roughly compares the different fluxes that may occur in the MS patients according to the number of enzymes downregulated and the enzymatic activity level of each one. Table 5 depicts, as a resume, upregulated and downregulated genes in distinct MS clinical forms and NMO and their related cumulative flux index values. 

With the calculation of CFI parameters, we may say that whole carbohydrate metabolic flux and ATP synthesis decreased in CGNs when exposed to CSF derived from MS and NMO patients. Our findings suggest a significant downregulation in the expression of genes involved in carbohydrate metabolism, suggesting that factors present in the CSF may perturb the metabolism of CGNs. 

## 4. Discussion

In the present study, we observed downregulated expression of genes involved in carbohydrate metabolism and ATP synthesis in CGNs exposed to CSF of MS patients as compared to controls. The expression of genes catalyzing essential steps of the glycolytic pathway, TCA cycle and oxidative phosphorylation were significantly reduced when CGNs were exposed to CSF of RMS subtype, particularly with the G+/M− clinical form. The total cumulative flux index (CFI), deduced from all respective pathways of carbohydrate metabolism, declined (CFI: 9.3 × 10^−4^) in neurons exposed to the CSF obtained from these patients. G+/M− MS is the least aggressive form of MS displaying oligoclonal bands of IgG antibodies but no IgM (G+/M−) oligoclonal antibodies are present in the CSF of these patients. Neuronal cells, as a result of neurological insult, are exposed to oxidative stress and require energy in the form of ATP to survive. However, downregulated expression of genes involved in glucose metabolism and consequently ATP production, as evidenced by our findings, resulted in bioenergetic failure in these CSF-treated neurons. In these patients, metabolic genes exhibited significantly reduced expression, leading to a reduction in the overall metabolic flux. This caused a decrease in ATP synthesis, which can be related to poor prognosis and strong development of pathology in these patients. Our results are consistent with previous findings, which demonstrated a reduction in ATP synthase expression in MS lesions [55].

Similarly, our data revealed downregulated expression of genes when neurons were exposed to CSF derived from patients with G+/M+ RMS. This is also a Relapsing MS subtype in which patients display oligoclonal bands of both IgG antibodies and IgM (G+/M+) antibodies in the CSF of brain. This form is more aggressive than G+/M− MS and patients have a better prognosis than the latter. The total cumulative flux index associated with glycolysis, TCA cycle and ATP generation declined to a great extent in this subtype of MS patients (CFI: 1.4 × 10^−5^). When those results are compared with those obtained of the G+/M− subtype, it may be indicated that expression of genes involved in metabolism of carbohydrates and ATP synthesis were most affected in the G+/M+ subtype. G+/M− RMS is a less aggressive, inflammatory clinical form of MS, with abundant oligoclonal IgG antibodies but no IgM antibodies in CSF. Thus, the energy required to repair damaged neurons degenerated in these MS lesions could be much lower than in the aggressive G+/M+ RMS. To combat oxidative stress generated in neurological diseases, cells need to produce large quantities of reducing equivalents for energy production and insufficient energy production severely impairs the ability to repair nerve damage. We conclude that the differential expression of metabolic genes influenced the bioenergetic profile of neurons, which could be related to worse prognosis in patients with RMS type G+/M + as compared to patients with RMS type G+/M−.

In the Spinal MS (Med), our findings revealed a significant reduction in the expression of genes implicated in glucose metabolism. Overall the accumulated total flux rate of glucose metabolism and ATP synthesis was found to be as low as 6.4 × 10^−5^ in these neurons. When data is compared with non-MS patients (controls), total cumulative flux index dropped to a very low value (CFI = 7.9 × 10^−7^) in all three metabolic pathways, similar to what occurred in G+/M+ RMS. This clinical form represents the most aggressive form of MS. All these patients were positive for oligoclonal IgG bands and negative for oligoclonal IgM bands in the CSF, suggesting that apart from the presence of OCMB in the CSF, there are other unidentified factors responsible of the aggressive course in MS. The drastic reduction of metabolic gene expression in both types of MS would result in bioenergetic failure, eventually causing oligodendrocytes to reduce the repair of axonal damage in neurons exposed to energy depletion of these aggressive types of MS, which is correlated with the worst prognosis compared with G+/M− RMS. 

Moreover, our findings revealed a reduced expression of genes involved in glucose metabolism in neurons treated with the CSF from PPMS patients. Altogether, total cumulative flux index in carbohydrates and ATP production was decreased in neurons (CFI = 2.7 × 10^−3^) in this subtype of patients. RMS is the most common form of MS, affecting 80–90% of the patients. On the contrary, patients with PPMS are characterized by a steady worsening of neurological symptoms with no relapse or remission, affecting 10–15% of the patients and lower progression time, suggesting that the metabolic affectation could be related to the duration period of the diseases.

Finally, there was an altogether different pattern of gene expression in neurons exposed to CSF of NMO patients than the MS types discussed above. We found that expression of genes that catalyze reactions in carbohydrate metabolism were overexpressed in neurons exposed to CSF derived from NMO patients. The overall total cumulative flux index in all three metabolic pathways remained virtually unchanged (CFI = 0.6779). The results indicate that the flux index accumulated in neurons exposed to the CSF ​​of NMO patients suffer a very slight decrease (CFI = 0.6779). Due to similar pathological characteristics exhibited by NMO, the disease was previously considered as a variant of MS. However, the identification of NMO specific IgG antibodies in the serum of NMO patients signifies that both diseases are different and require a different therapeutic approach for treatment. Unlike MS, NMO is not a progressive disease. In fact, less than 2% of patients have progression leading to disability. This clinical difference alters the gene expression in neurons exposed to CSF derived from NMO patients and hold on the hypothesis that the disturbance of the glucose metabolism occurs in the settings (or are responsible of) the progressive increase of the neuroaxonal destruction.

Several studies have demonstrated the unmet need of energy requirement in MS. This results in mitochondrial impairment in cultured neurons [56], animal models of MS [57] and in MS samples [58]. Metabolic abnormalities are implicated in the pathogenesis of neurodegenerative diseases [12,13,16,18,19,20,59]. Evidence in the MS literature indicating any association between perturbed glucose metabolism with its pathogenesis is meagre. Some investigators have determined the activities of metabolic enzymes such as enolase, pyruvate kinase, lactate dehydrogenase and aldolase in the CSF of MS patients and they have found increased levels [14]. Moreover, there is an intrathecal production of B cells and antibodies against *Tpi* and *Gapdh* genes in the CSF and lesions of MS patients [60]. These antibodies suppress the glycolytic activity of GAPDH but not TPI in MS patients [15]. 

The main findings of this study revealed a disturbed carbohydrate metabolism in CGNs treated with CSF derived from MS and NMO subjects. Factors present in the CSF, in our model, affected the metabolism of neurons and clearly differentiate more benign forms from the most aggressive forms of MS. CSF factors from MS caused death in neurons but not in astrocytes. The effect of CSF was different in MS aggressivity of RMS, (G+/M−, G+/M+), Spinal MS and PPMS clinical forms. G+/M+ and Spinal MS derived CSF-treated neurons and OPCs were strongly affected by reducing carbohydrate metabolism, as evidenced by downregulation of expression of most of the genes investigated, which is suggestive of reduced ATP synthesis. This indicates more damage to neurons, which is correlated with worst prognosis. The NMO-derived CSF-exposed neurons revealed increased carbohydrate metabolism, as indicated by increased expression of genes, whereas neurons treated with CSF from G+/M− MS patients demonstrated slightly reduced carbohydrate metabolism with less neuronal damage correlated with poor prognosis. 

These results allow us to differentiate different clinical forms and aggressivity in MS and to differentiate MS from NMO. However, whether these alterations in metabolic gene expression cause MS and NMO or are simply a mere consequence of the diseases remains unknown. These findings open new avenues of study and allow the development of therapeutic agents targeted to restore the metabolic function and hence repair and/or prevent axonal damage responsible for functional disability in the patient. A greater understanding of these impaired metabolic pathways may offer new insights into more efficacious treatments for both MS and NMO.

## 5. Conclusions

The data showed in this work indicate that factors present in CSF factors from MS cause selectively death in neurons but not astrocytes. These factors induce an alteration in glucose metabolism and ATP production that seems to be related with neuronal death. 

The CSF effect is clearly different according with MS aggressivity, higher in RMS and PPMS clinical form. In G+/M+ RMS and spinal MS (medullary) derived CSF, the treated neurons are strongly affected by reducing carbohydrate metabolism as evidenced by down regulation of most of the genes, which is suggestive of least ATP synthesis capacity. This indicates more damage to neurons and correlated with worst prognosis. The NMO derived CSF exposed neurons reveals an increased carbohydrate metabolism as indicated by increased expression of genes whereas neurons treated with CSF from G+/M− RMS patients demonstrates slightly reduced carbohydrate metabolism with less neuronal damage and correlate with poor prognosis of these patients. 

In conclusion these findings indicate that some factors present in the CSF are affecting, in our model, the carbohydrate metabolism and ATP synthesis of neurons and clearly differentiate more benign forms from the most aggressive forms in MS. The study also differentiates NMO from MS, which is sometimes difficult to distinguish by the clinicians.

## Figures and Tables

**Figure 1 brainsci-08-00001-f001:**
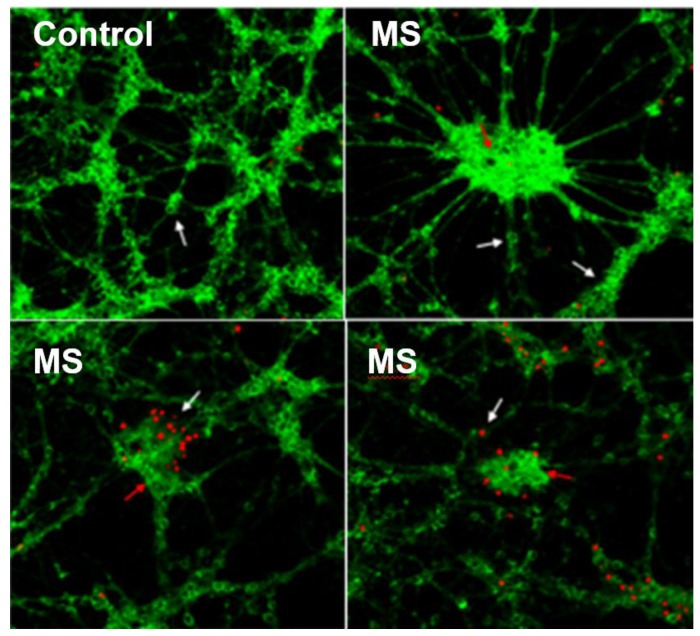
Viability and cell death in cerebellar granule neurons (CGN) (white arrow) and astrocytes (red arrow). Fourteen-days culture of CGNs were treated with cerebrospinal fluid (CSF) of Multiple Sclerosis (MS) patients and stained with propidium iodide (red fluorescence) and rhodamine-123 (green fluorescence). The images correspond with three independent experiments.

**Figure 2 brainsci-08-00001-f002:**
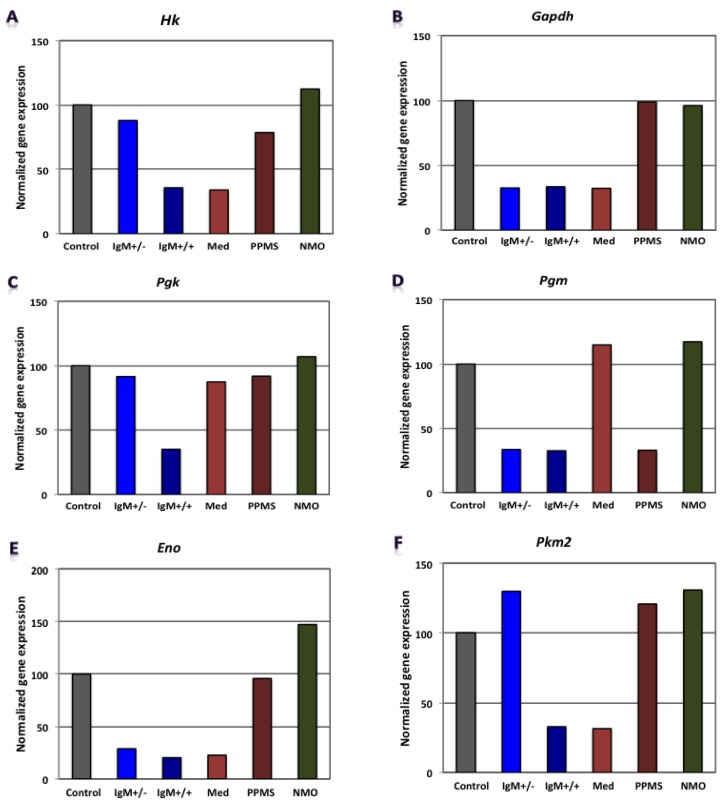
Normalized gene expression involved in glycolytic pathway in CGNs treated with CSF of MS and Neuromyelitis optica (NMO) patients. (**A**) Hexokinase Hk; (**B**) Glyceraldehyde-3-phosphate dehydrogenase, Gadph; (**C**) Phosphoglycerate kinase, Pgk; (**D**) Phosphoglycerate mutase, Pgm; (**E**) Enolase, Eno; (**F**) Pyruvate kinase isozyme M2, Pkm2. CGNs: cerebellar granule neurons; RMS (G+/M− and G+/M+): types of inflammatory MS; SMS: Spinal—MS (Med); PP: Primary progressive multiple sclerosis; NMO: Neuromyelitis Optica.

**Figure 3 brainsci-08-00001-f003:**
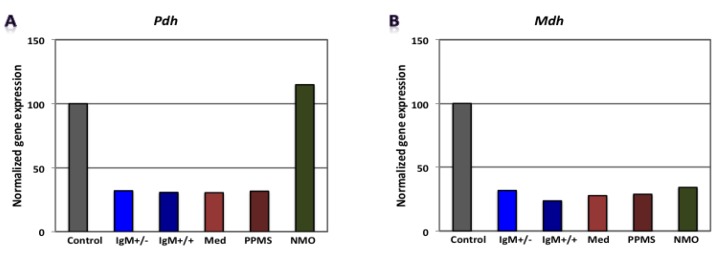
Normalized gene expression involved in tricarboxylic acid (TCA) cycle in CGNs treated with CSF of MS and NMO patients. (**A**) Pyruvate dehydrogenase, Pdh; (**B**) Malate dehydrogenase, Mdh. CGNs: cerebellar granule neurons; RMS (G+/M− and G+/M+): types of inflammatory MS; Spinal MS (Med); PP: Primary progressive multiple sclerosis; NMO: Neuromyelitis Optica.

**Figure 4 brainsci-08-00001-f004:**
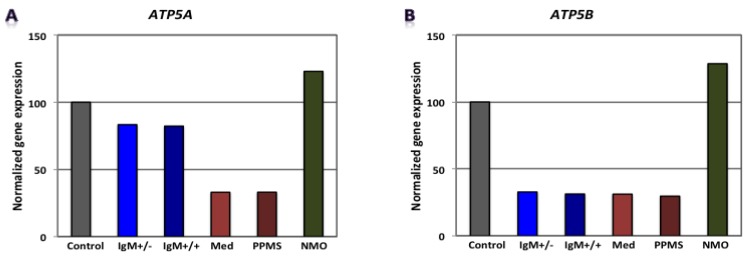
Normalized gene expression involved in oxidative phosphorylation in CGNs treated with CSF of MS and NMO patients. (**A**) ATP synthase subunit 5A, ATP5A; (**B**) ATP synthase subunit 5B, ATP5B. CGNs: cerebellar granule neurons; RMS (G+/M− and G+/M+): types of inflammatory MS; SMS: Spinal MS (Med); PP: Primary progressive multiple sclerosis; NMO: Neuromyelitis Optica.

**Figure 5 brainsci-08-00001-f005:**
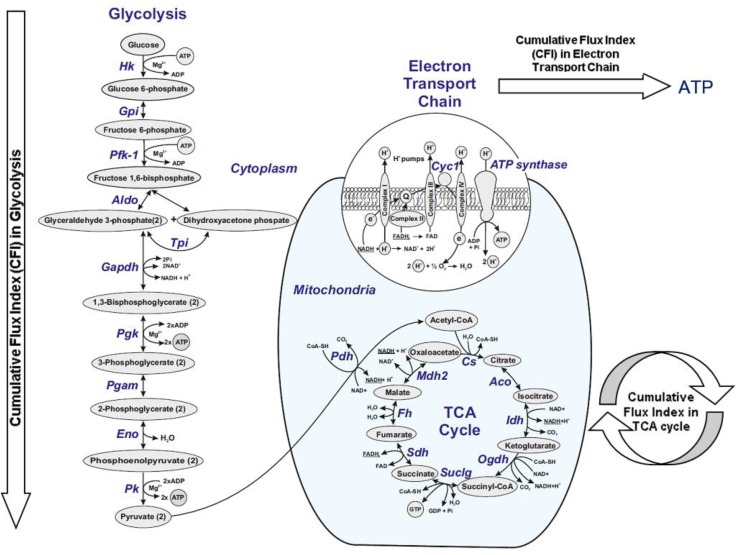
A schematic diagram of a general metabolic network including glycolysis, tricarboxylic acid (TCA) cycle and electron transport chain. The cumulative flux index (CFI) values were calculated as a parameter to integrate the reduced activity of the different enzymes in a specific network (glycolysis, TCA cycle and ATP generation by Electron Transport Chain (ETC), or together) as the expected total flux.

**Figure 6 brainsci-08-00001-f006:**
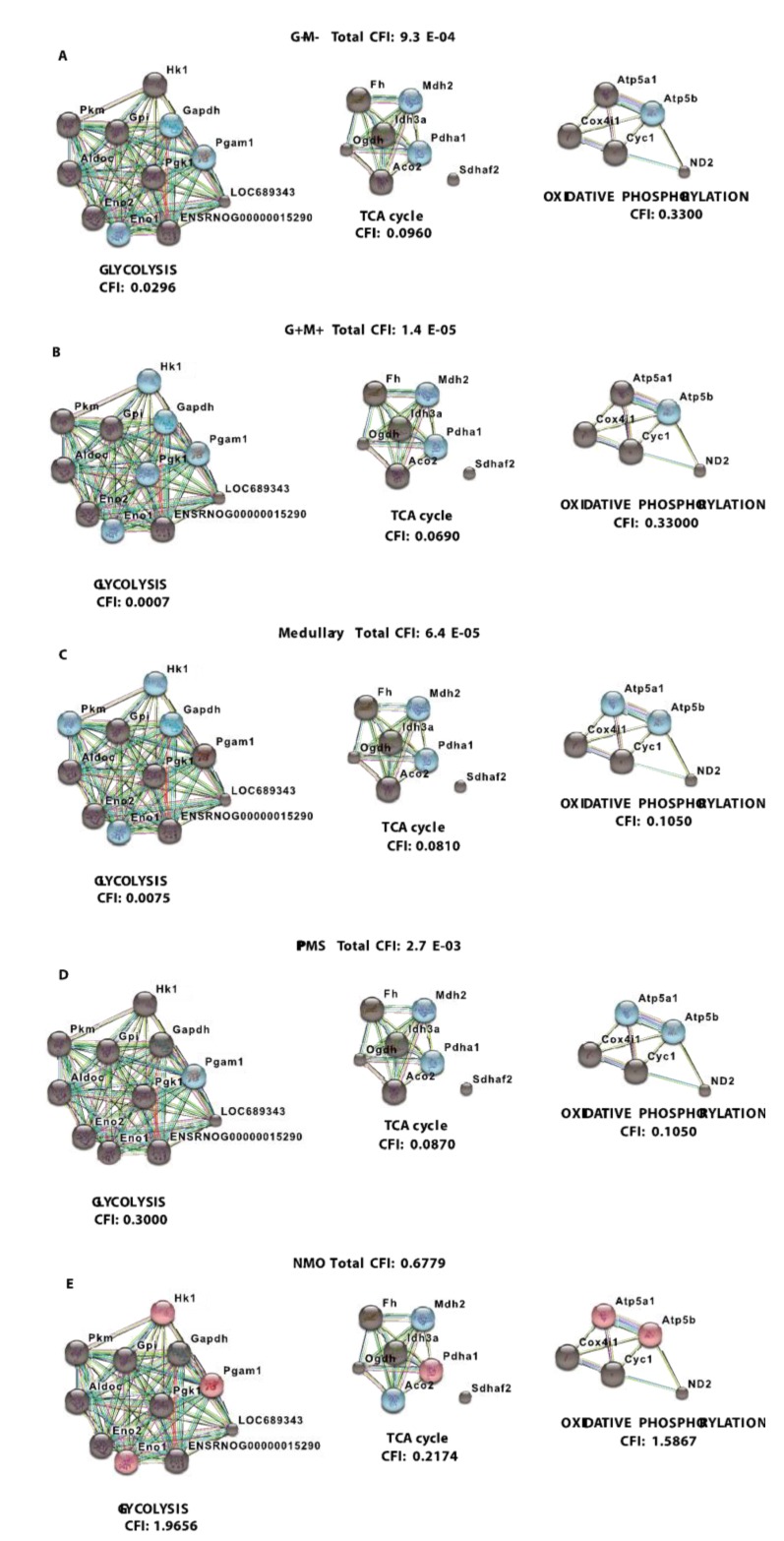
Visualization of gene interaction network generated by STRING v10. Visualization of gene interaction network generated by STRING v10 in CGNs cultures exposed to CSF from patients with (**A**) G+/M− RMS; (**B**) G+/M+ RMS; (**C**) Spinal MS (Med); (**D**) PPMS; (**E**) NMO. Different line colors represent the types of evidence for the association between genes involved in glycolysis, TCA cycle and oxidative phosphorylation in neurons; Red color signifies upregulated expression, blue color signifies downregulated expression and grey color signifies genes with no variation in expression in our experimental conditions. LOC689343 signifies the *Pk* gene and ENSRNOG00000015290 (a specific nomenclature gave by the program) signifies the *Tpi* gene.

**Table 1 brainsci-08-00001-t001:** Clinical characteristics of the patients. Case = Patient number; G = Gender; F = Female; M = Male; Age = patient age in years; P.T. = progression time (duration of the disease since diagnosis in years); Actual EDSS = Expanded disability status scale; OCGB IgG = oligoclonal IgG bands in CSF; OCMB IgM = oligoclonal IgM bands in CSF; Ac-AQ4 = Antiaquaporin 4 antibodies; T = Treatment. Notations: RMS = Relapsing multiple sclerosis; SPMS = Secondary progressive multiple sclerosis; CIS = Clinically isolated syndrome; Spinal MS = Medullary MS; PPMS = Primary progressive multiple sclerosis; NMO = Neuromyelitis optica patients; G+/M− = presence of IgG but no IgM antibodies in the CSF; G+/M+ = presence of both IgG and IgM antibodies in the CSF; N.A. = not applicable; N.T. = no treatment at the time of collection of the samples; P. = presence of Ac-AQ4; N. = non-detected Ac-AQ4; IFN = interferon; FGM = fingolimod; MTZ = mitozantrone; U. = unavailable ; CPX = copaxone; NTZ = natalizumab; IV-IgG = bi-monthly pulsed intravenous immunoglobulin; PE = plasma-exchange 2 months before lumbar puncture; ASCT = autologous stem cell transplant one year before LP; Cy = Cyclophosphamide; RTX = rituximab five months before lumbar puncture.

Case	G	Age	Working Clinical Form	Clinical Form	P.T.	Actual EDSS	OCGB IgG	OCMB IgM	Ac-AQ4	T
**Multiple Sclerosis**										
1	F	23	RMS (G+/M−)	RRMS	5	1.50	G+	M-	N.A.	IFN, FGM
2	F	21	RMS (G+/M−)	SPMS	18	4.00	G+	M-	N.A.	IFN, MTZ
3	F	36	RMS (G+/M−)	CIS	4	1.50	G+	M-	N.A.	N.T.
4	M	22	RMS (G+/M−)	RRMS	6	1.50	G+	M-	N.A.	IFN, CPX, NTZ, FGM
5	F	21	RMS (G+/M−)	RRMS	3	3.00	G+	M-	N.A.	IFN
6	F	30	RMS (G+/M−)	RRMS	22	4.00	G+	M-	N.A.	IFN, NTZ, FGM
7	F	29	RMS (G+/M−)	RRMS	10	1.50	G+	M-	N.A.	N.T.
8	F	29	RMS (G+/M−)	RRMS	7	1.50	G+	M-	N.A.	IFN, NTZ
9	F	28	RMS (G+/M−)	RRMS	10	5.50	G+	M-	N.A.	MTZ, IFN
10	F	28	RMS (G+/M−)	RRMS	4	1.00	G+	M-	N.A.	N.T.
11	F	37	RMS (G+/M+)	RRMS	7	3.50	G++	M+	N.A.	IFN, NTZ, CPX
12	M	32	RMS (G+/M+)	RRMS	4	1.00	G+	M+	N.A.	IV-IgG
13	F	44	RMS (G+/M+)	RRMS	5	2.00	G+	M+	N.A.	N.T.
14	F	26	RMS (G+/M+)	RRMS	5	2.00	G++	M++	N.A.	U.
15	F	14	RMS (G+/M+)	RRMS	18	3.50	G+	M+	N.A.	PE, IFN, NTZ, ASCT
16	M	25	RMS (G+/M+)	RRMS	11	2.00	G+	M+	N.A.	IFN, FGM
17	F	21	RMS (G+/M+)	SPMS	25	8.50	G++	M+	N.A.	IFN, AZA, MTZ, IV-IgG
18	F	17	RMS (G+/M+)	RRMS	16	2.00	G+	M+	N.A.	U.
19	F	23	RMS (G+/M+)	SPMS	18	6.50	G+	M+	N.A.	IFN, IV-IgG, Cy
20	F	22	RMS (G+/M+)	SPMS	5	4.00	G+	M+	N.A.	IFN, FGM, CPX
21	F	29	RMS (G+/M+)	RRMS	5	2.50	G+	M+	N.A.	IFN, FGM, CPX
22	M	39	Spinal MS	SPMS	10	4.50	G+	M+	N.A.	IFN, Cy, FGM, NTZ
23	F	25	Spinal MS	SPMS	6	7.00	G+	M-	N.A.	IFN, MTZ, RTX
24	F	25	Spinal MS	SPMS	14	8.00	G+	M+	N.A.	IFN, Cy, RTX
25	M	34	Spinal MS	SPMS	9	6.00	G+	M-	N.A.	IFN, Cy
26	M	34	Spinal MS	SPMS	6	6.50	G+	N.A.	N.A.	IFN, MTZ, IV-IgG
27	F	23	Spinal MS	RRMS	5	4.00	G-	M-	N.A.	IFN, NTZ, FGM, RTX
28	F	40	Spinal MS	SPMS	10	7.50	G-	M+	N.A.	AZA, IV-IgG, Cy, RTX
29	F	23	Spinal MS	SPMS	28	6.50	G+	M-	N.A.	IFN, MTZ, RTX
30	F	54	PPMS	PPMS	12	7.00	G+	M-	N.A.	N.T.
31	M	40	PPMS	PPMS	23	6.00	G+	M-	N.A.	MTZ, Cy, RTX
32	F	52	PPMS	PPMS	14	5.50	G++	M-	N.A.	AZA
33	F	38	PPMS	PPMS	11	5.50	G+	M-	N.A.	N.T.
34	M	31	PPMS	PPMS	24	6.00	G+	M-	N.A.	N.T.
35	F	47	PPMS	PPMS	14	5.50	G++	M-	N.A.	N.T.
36	M	49	PPMS	PPMS	11	6.00	G-	M-	N.A.	FGM
37	F	26	PPMS	PPMS	13	6.50	G++	M-	N.A.	N.T.
38	F	34	PPMS	PPMS	6	5.00	G++	M-	N.A.	N.T.
39	F	39	PPMS	PPMS	8	8.50	G-	M-	N.A.	Cy
40	M	18	PPMS	PPMS	15	8.00	U.	U.	N.A.	U.
**Neuromyelitis Optica Patients**										
41	F	39	NMO	NMO	5	9.00	G-	M-	P.	IFN, MTZ, Cy, RTX
42	F	50	NMO	NMO	4	7.00	G-	M-	P.	IFN, NTZ, RTX
43	M	15	NMO	NMO	17	4.00	G+	M+	N.	IFN, IV-IgG
44	M	42	NMO	NMO	5	3.50	N.A.	N.A.	P.	IV-IgG
45	F	22	NMO	NMO	5	2.50	G+	M+	P.	IV-IgG, IFN, CPX
46	F	27	NMO	NMO	5	2.00	G+	M-	N.	IFN, AZA
47	M	9	NMO	NMO	14	1.00	G-	M-	N.	IV-IgG, IFN, CPX
48	F	8	NMO	NMO	32	4.00	G+	M+	N.	IFN, IV-IgG
49	M	19	NMO	NMO	20	8.50	G-	M-	N.	IFN, MTZ, Cy, RTX
**Non-Inflammatory Neurological Diseases (Control)**										
50	M	23	CONTROL	Non-inflammatory motor neuron neurological disease	N.A.	N.A.	G-	M-	N.A.	N.T.
51	F	77	CONTROL	Glioblastoma	N.A.	N.A.	G-	M-	N.A.	N.T.
52	F	33	CONTROL	Central Pontine myelinolisis	N.A.	N.A.	G-	M-	N.A.	N.T.
53	F	32	CONTROL	Central Pontine myelinolisis	N.A.	N.A.	G-	M-	N.A.	N.T.
54	M	59	CONTROL	Classic migraine	N.A.	N.A.	G-	M-	N.A.	N.T.
55	F	36	CONTROL	Headache in which subarachnoid hemorrhage was suspected	N.A.	N.A.	G-	M-	N.A.	N.T.
56	F	57	CONTROL	Headache in which subarachnoid hemorrhage was suspected	N.A.	N.A.	G-	M-	N.A.	N.T.
57	M	37	CONTROL	Headache in which subarachnoid hemorrhage was suspected	N.A.	N.A.	G-	M-	N.A.	N.T.
58	F	21	CONTROL	Benign intracranial hypertension	N.A.	N.A.	G-	M-	N.A.	N.T.
59	M	13	CONTROL	Chronic axonal polyneuropathy	N.A.	N.A.	G-	M-	N.A.	N.T.

**Table 2 brainsci-08-00001-t002:** General characteristics of series studied. SD: standard deviation; NA: not applicable; EDSS: expanded disability status scale.

	Controls (*n =* 10)	MS Patients (*n =* 40)	NMO Patients (*n =* 9)	*p*
% females (*n*)	60.0 (6)	75.0 (30)	55.6 (5)	0.40 (*χ*^2^)
Age (mean, SD)	40.3 (19.5)	30.7 (9.7)	25.6 (15.0)	0.04 (ANOVA test)
EDSS	NA	4.5 (2.3)	4.6 (2.8)	0.94 (*t*-test)
Progression Time	NA	11.1 (6.6)	11.8 (9.7)	0.79 (*t*-test)

**Table 3 brainsci-08-00001-t003:** Characteristics of MS patients according to clinical classification. SD: standard deviation; EDSS: expanded disability status scale.

	RMS (*n =* 18)	SPMS (*n =* 11)	PPMS (*n =* 11)	*p*
% females (*n*)	83.3 (15)	72.7 (8)	63.6 (7)	0.48 (*χ*^2^)
Age (mean, SD)	27.3 (7.2)	27.9 (7.3)	38.9 (11.2)	0.003 (ANOVA test)
EDSS	2.4 (1.2)	6.2 (1.5)	6.3 (1.1)	<0.001 (ANOVA test)
Progression Time	8.1 (5.4)	13.5 (7.8)	13.8 (5.7)	0.043 (ANOVA test)

**Table 4 brainsci-08-00001-t004:** Characteristics of MS patients according to new proposal and working classification. SD: standard deviation; NA: not applicable; EDSS: expanded disability status scale; MSSS: multiple sclerosis severity score.

	Relapsing MS (*n =* 21) G+/M− (*n =* 10) G+/M+ (*n =* 11)		Spinal MS (*n =* 8)	PPMS (*n =* 11)	*p*
% females (*n*)	90 (9)	81.8 (9)	62.5 (5)	63.6 (7)	0.40 (*χ*^2^)
Age (mean, SD)	26.7 (4.8)	26.3 (8.7)	31.4 (7.0)	38.9 (11.2)	0.005
EDSS	2.5 (1.5)	3.4 (2.2)	6.2 (1.4)	6.3 (1.1)	<0.001
MSSS *	3.2 (1.9)	4.5 (2.5)	8.2 (1.6)	6.9 (1.6)	<0.001
Progression Time	8.9 (6.3)	10.8 (5.9)	8.5 (3.1)	13.8 (5.7)	0.154

**Table 5 brainsci-08-00001-t005:** Differentially expressed genes, which are upregulated and downregulated in distinct MS clinical forms and NMO and their related cumulative flux index. TCA: tricarboxylic acid cycle; ETC: Electron Transport Chain; CFI: Cumulative Flux Index.

Clinical Form	Genes Upregulated			Genes Downregulated			CFI			Total CFI	Prognosis
	Glycolysis	TCA Cycle	ETC	Glycolysis	TCA Cycle	ETC	Glycolysis	TCA Cycle	ETC		
RMS (G+/M−)	*-*	*-*	*-*	*Gapdh*, *Pgam1*, *Eno1*	*Mdh2*, *Pdha1*	*Atp5b*	0.0296	0.0960	0.3300	9.3 × 10^−4^	Poor prognosis
RMS (G+/M+)	*-*	*-*	*-*	*Hk1*, *Gapdh*, *Pgam1*, *Pgk1*, *Eno1*	*Mdh2*, *Pdha1*	*Atp5b*	0.0007	0.0690	0.33000	1.4 × 10^−5^	Worse prognosis
Spinal MS	*-*	*-*	*-*	*Hk1*, *Gapdh*, *Eno1*, *Pk1*	*Mdh2*, *Pdha1*	*Atp5a1*, *Atp5b*	0.0075	0.0810	0.1050	6.4 × 10^−5^	Worst prognosis
PPMS	*-*	*-*	*-*	*Pgam1*	*Mdh2*, *Pdha1*	*Atp5a1*, *Atp5b*	0.3000	0.0870	0.1050	2.7 × 10^−3^	-
NMO	*Hk1*, *Pgam1*, *Eno1*	*Pdha1*	*Atp5a1*, *Atp5b*	*-*	*Mdh2*, *Aco2*	*-*	1.9656	0.2174	1.5867	0.6779	-

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
