# Peer review of "Disturbed Glucose Metabolism in Rat Neurons Exposed to Cerebrospinal Fluid Obtained from Multiple Sclerosis Subjects"

_brainsci, 2017, doi:10.3390/brainsci8010001_

Round 1

Reviewer 1 Report

The paper investigates the correlation between disturbed glucose metabolism and neurodegenerative diseases.

It is known that metabolic abnormalities are implicated in the pathogenesis of neurodegenerative diseases, but the MS literature indicating any association between perturbed glucose metabolism with its pathogenesis is still poor.
For this, the subject is timely and the results are interesting, also for future clinical implications.
A greater understanding of these metabolic pathways may offer new insights for more efficacious treatments for this deseases, in particular for NMO.
The paper is well written and has been adequately discussed
However, it is required to confirm and integrate these preliminary results with other studies.

Author Response

Thanks for the comment. We have rewritten the manuscript and have tried to better integrate our results with the literature related with MS and metabolism.

Reviewer 2 Report

The aim of this study was to investigate if there are any substances with cellular toxicity in CSF from MS patients, MNO patients and controls. The study design was to add CSF to cerebellar granule neurons (CGN) from the forebrain of rats in culture dishes. Gene expression and confocal microscopy were used to detect differences. The main finding is that the carbohydrate metabolism was disturbed by CSF from MS patients.

I do not approve with the design of the study using the controls are normalizing for gene expression. One must first find a house hold gene that are expressed equally in all the samples, normalize to this gene and then normalize to the controls. This is to show your credentials of being able to culture cells in the same way in all your experiments.

In addition the quality of the language and writing style need substantial improvements highlighting the essential message, reducing redundancies and presenting the data in a clear way. 

Author Response

Thanks for the comment. We are sorry we did not explain clearly the methodology used for both normalizations. We explained in materials and methods, and in results, how we selected the housekeeping genes (also described in our paper Mathur et al., 2015) to normalize data of microarrays for the genes related with carbohydrate metabolism when compared treatment CSF treatment of different MS and NMO patients.

See pg. 8-9 of the revised manuscript: “Differentially expressed genes were identified by comparing average expression levels in cases and controls. To normalize the data of expression of genes related with the carbohydrate metabolism, we selected a panel of reference genes from microarray data and analyzed by geNorm and NormFinder algorithms. After the analysis, we find that Tfrc and B2m the most stably expressed genes in the treatment with the CSF of the different MS and NMO patients (Mathur et al., 2015). Those genes were used as endogenous controls to normalize the microarray data obtained.

After this normalization, the mRNA expression (in terms of absolute fold change) in neurons treated with CSF of MS and NMO individual patients was compared with gene expression in neurons exposed to CSF from neurological controls. Fold change cut off was considered as 2. The microarrays data correspond to at least 3-4 independent assay per group of individual MS patients and controls”.

See pg. 12 of the revised manuscript: “The genes related with carbohydrate metabolism were normalized using as endogenous controls expression signal of Tfrc and B2m genes, two stably expressed genes in the treatment with the CSF of the different MS and NMO patients (Mathur et al., 2015)”.

We apologize for English language mistakes. The revised version was checked by a professional in writing English to reduce those mistakes in the manuscript. The manuscript was also rewritten trying to show the results and discussion in a clearer approach for better reading.

Round 2

Reviewer 2 Report

Better, but still unclear text that needs editing. The figure text of figure 6 A-E gives description of  fig 6 A-L, which does not make sense. The concept of evolution of disease is unusual and I wonder if you might replace it with disease duration if this is what you mean. 

Author Response

Thanks for the comments. We have made the changes in the revised manuscript.
